# SIW-Based Circularly Polarized Antenna Array for 60 GHz 5G Band: Feasibility Study

**DOI:** 10.3390/s22082945

**Published:** 2022-04-12

**Authors:** Jan Spurek, Zbynek Raida

**Affiliations:** Faculty of Electrical Engineering and Communication, Brno University of Technology, Technicka 12, 616 00 Brno, Czech Republic; janspurek@gmail.com

**Keywords:** substrate-integrated waveguide (SIW), antenna array, circular polarization, parasitic patches, conversion

## Abstract

At present, most millimeter wave 5G systems operate at frequencies ranging from 24 GHz to 39 GHz. Nevertheless, the new 5G release is going to increase the supported 5G spectrum into the 60 GHz band. In this communication, we discuss a methodology of converting a modular antenna array, which was originally designed for the 17 GHz unlicensed band, to the emerging 60 GHz 5G band. The antenna array is of a modular architecture, allowing a simple extension to a higher number of elements. The feeding structure is composed of substrate-integrated waveguides (SIW), allowing relatively simple and cheap manufacturing. As revealed by a sensitivity analysis, the frequency up-conversion significantly increases the need for precision of the used manufacturing technology. Subsequently, the structure is optimized to minimize the repercussions of manufacturing variations. The properties of the converted array are studied when equipped with parasitic patches to increase the axial ratio (AR) bandwidth.

## 1. Introduction

For more than a decade, the potential exploitation of the 60 GHz frequency band for 5G communication has been discussed. The 5G networks aim to utilize the 60 GHz ISM band mainly for point-to-point links with the objective to increase throughput by the bandwidth expansion [1,2,3]. Hence, the demand for new antenna structures able to satisfy the 5G 60 GHz requirements is increasing.

Circularly polarized antennas and arrays are conveniently used in applications where mutual orientation of the transmitter and the receiver is arbitrary in time. To address such needs over a wide range of frequencies, a circularly polarized antenna array for the 17 GHz unlicensed band was designed in [4]. When converting the design to the emerging 60 GHz 5G band, the influence of the design solution on the properties of the array was investigated. A modular antenna array solution was introduced, from which a variety of use cases can benefit.

The requirements on the designed 5G 60 GHz antenna array follow the specifications defined in [1]. Within the frequency range from 57 to 66 GHz, impedance matching should meet the condition |S11| < −10 dB, the axial ratio AR < 3 dB should ensure the circular polarization, and the realized gain should exceed 10 dBi.

Let us turn our attention to the available concepts of circularly polarized antenna arrays, which were designed for the 60 GHz 5G band and were excited by substrate-integrated waveguides (SIW).

An array consisting of rectangular patches with truncated corners, which were excited via apertures and fed by a SIW network, was presented in [5]. In [6], patches with truncated corners were replaced by rotated patches with a rhombic slot in the center, and a linear aperture was replaced by an L-shaped slot. Unfortunately, only outputs of simulations were provided for both the array concepts.

In [7], an array consisting of circular patches was excited by rectangular SIW resonators. Compared to the presented design, an additional layer in the feeding structure is requested, and other array concepts differ from [7] marginally:Circular patches were replaced by spirals in [8].Linear apertures were replaced by crosses and elements by rotated cross-shaped patches [9].In [10], three additional feeding layers were used, making the fabrication highly demanding. In the paper, no information about the axial ratio was provided.In [11], phase shifters were used so that the circular polarization could be excited. The implementation of phase shifters increases the antenna dimensions.

In [12], septum-based antenna elements are exploited, creating a leaky-wave antenna. The concept of this structure is difficult to compare with the presented array. None of the reviewed concepts provides such a level of modularity as the presented array, and none of the concepts apply parasitic patches for the improvement of circular polarization.

## 2. From 17 GHz to 60 GHz

The original array was composed of two layers of the ARLON CuClad 217LX substrate with the thickness *t* = 1.508 mm and the dielectric constant *ε_r_* = 2.17. The array consisted of four patches on the top layer, and those served as radiating elements. To radiators, the signal was delivered in-phase by a waveguide network integrated in both the substrates. Employing power dividers, phase inverters, and coupling slots, the transition between the substrate layers was provided, and the patches were excited.

Signal paths within the antenna array are shown in Figure 1. From the input port in the top layer, the divided power is forwarded to the bottom layer via two slots. In the bottom layer, the power is again divided. In each branch, two patches are excited via apertures.

Implementation of the antenna array was based on substrate-integrated waveguides. As discussed in Section 3, an implementation with smaller vias was replaced by larger vias to allow the manufacturing in a university lab with lower production accuracies. Positions of larger vias in the top substrate and the bottom substrate are depicted in Figure 2.

The array was designed to radiate a left-handed circularly polarized wave (LHCP) with the main lobe directed perpendicularly to the surface of the substrate. The substrate-integrated network was fed by an SMA to the grounded coplanar waveguide (GCPW) transition.

When converting the array to a higher frequency, the dimensions were proportionally reduced, as dictated by the shortened wavelength of the signal and principles the array operates on. The used substrate was preserved with the thickness reduced to 0.508 mm. As indicated in the datasheet, the dielectric constant and the dissipation factor of ARLON CuClad 217 showed the stability when moving towards higher frequencies [13]. In order to verify an influence of worsening substrate parameters with the frequency, we considered the shift of the dielectric constant *ε_r_* = 2.17 → 2.18 and the shift of the loss tangent tan *δ* = 0.001 → 0.002 and simulated the array with these values. The impedance characteristics moved to higher frequencies for 0.2 GHz approximately, the polarization characteristic moved to higher frequencies for 0.1 GHz approximately, and the radiation patterns showed a slight decrease of realized gain. This effect seems to be marginal.

As a prerequisite prior manufacturing, a sensitivity analysis of the modified model was performed to assess the allowed tolerances with regards to the manufacturing technology. Moreover, a new feeding interface in the form of a rectangular waveguide to substrate-integrated waveguide (SIW) transition was designed, since the original SMA to GCPW transition used at 17 GHz was no longer suitable [14].

The sensitivity analysis identified the following parameters as critical, since most influenced the performance and properties of the array:Dimensions of coupling slots between the layers;Position of coupling slots between the layers;Width of the waveguides;Dimensions of the patches.

These parameters put most of the demand on the correct alignment of two substrate layers, the precision of drilling the vias and the precision of etching used to create the patches.

## 3. Results

In the following paragraphs, the measured parameters of the implemented antenna arrays are compared with the simulated results.

In order to model the antennas, a transient solver of CST Microwave Studio was used. The solver is based on the implementation of the finite integration technique [15], which is a genuine 3D numerical method based on the full-wave formulation of Maxwell equations and a staircase discretization mesh.

In the initial step, side walls of substrate-integrated waveguides were assumed to be solid. Moreover, perfect electric conductivity of all metallic surfaces was considered, and lossless dielectrics were expected to reduce the simulation time. Thanks to the simulation of this idealized geometry, excitation of the spurious modes was eliminated.

At the advanced stage of the design, the ideal properties of the materials were changed to realistic models, and solid side walls of the waveguides were replaced by vias. No symmetries were considered, and the complete structure was simulated.

The simulated antennas were manufactured by etching in the university lab with a maximum accuracy 0.2 mm.

The manufactured antennas were measured by the vector network analyzer Rohde&Schwarz ZVA110. The radiation properties of the antennas were characterized by the spherical antenna measurement systems NSI 700S-30 and NSI 700S-90E. The scanners, both installed in an anechoic chamber, were able to measure the radiation properties of the antennas in the frequency range from 800 MHz to 50 GHz in the near-field range and up to 110 GHz in the far-field range.

### 3.1. Smaller Vias

The first prototype employed vias with a smaller diameter (0.2 mm), with the goal of obtaining better properties of the SIW power delivery network, is depicted in Figure 3.

Figure 4 compares the simulated and measured frequency responses of the reflection coefficient. Obviously, the impedance bandwidth of the measured array was smaller (59.72–62.86 GHz, 5.2% relatively) in comparison to the simulated array (58.9–62.4 GHz, 5.8% relatively). In addition, the characteristic was shifted towards a higher frequency.

Only a minimal match between the simulated and measured frequency responses of the axial ratio was identified between 58 GHz and 59 GHz and between 62 GHz and 63 GHz, where the measured characteristic reaches its minimum magnitude. The |AR| < 3 dB bandwidth of the simulated array is from circa 57.5–62.5 GHz (8.3% relatively), and the bandwidth of the measured array is only from 58.5–58.9 GHz (0.7% relatively) and from 62.3 GHz to 62.5 GHz (0.3% relatively).

The reflection coefficient and axial ratio (AR) characteristics show that this prototype does not operate as expected. Manufacturing defects of the vias that form the SIW were identified as the cause: their diameter was relatively small (0.2 mm), and the ratio between the diameter and the height of the vias was beyond the recommended limit. Hence, the array utilizing vias with a larger diameter was investigated to allow more reliable manufacturing.

### 3.2. Larger Vias

In order to lower the chance of the defects associated with manufacturing, the diameter of the vias was increased from 0.2 mm to 0.6 mm, as illustrated in Figure 5.

Figure 6 compares reflection coefficients for the simulated arrays with smaller and larger vias. Increasing the diameter of the vias, the impedance bandwidth is increased due to the lower resonance shift towards lower frequencies: 58.9–62.4 GHz (5.8% relatively) for the array with smaller vias and 57.6–62.4 GHz (8% relatively) for the array with larger ones.

Figure 7 compares the AR for the simulated arrays with smaller and larger vias. Obviously, the |AR| < 3 dB bandwidth of the array with larger vias was shifted towards the lower frequencies by approximately 1 GHz: 57.5–62.5 GHz (8.3% relatively) for the array with the smaller vias and 56.5–61.5 GHz (8.3% relatively) for the array with the larger vias.

For both versions of the array, the same WR15 to SIW transition was used (see Figure 8). The transition from a classical WR15 waveguide was first realized by eight vertical steps to gradually assume the height of the SIW, and then, a linear horizontal taper was used to match the SIW width. The outer dimensions of the transition were chosen with respect to the standardized waveguide equipment in order to attach it to the measurement instruments.

For the simulated arrays with smaller and larger vias, Figure 9 compares radiation patterns in the *xy* plane (top) and the *yz* plane (bottom). Obviously, the array with larger vias showed a higher realized gain (13.3 dB vs. 12.2 dB in the main lobe direction) and a more uniform pattern in the *yz* plane. The beam width was reduced in both planes in the case of the array with larger vias: 30.9° vs. 22° in the *xy* plane and 28.8° vs. 23° in the *yz* plane.

Based on the comparisons, the changes in the properties of the array were evaluated as acceptable, and the array was manufactured; the manufactured array with larger vias is shown in Figure 10. Here, 3D-printed elements (black) were designed and manufactured to align and fix two substrate layers of the array more precisely. For 3D printing, a PETG filament was used.

Figure 11 compares the simulated and measured frequency responses of the reflection coefficient of the array with larger vias. Obviously, a better match with the simulations was obtained compared to the array with the smaller via diameter. By using the data obtained by the sensitivity analysis [14], we identified that deviations from the ideal dimensions of the coupling slots between the substrate layers and/or misalignment of the substrate layers in the manufactured array are likely responsible for the differences. The characteristic is consistent with the situation where the length of the coupling slot is increased by 0.05 mm and the width of the slot is decreased by 0.075 mm. The impedance bandwidth of the simulated array is 57.6–62.4 GHz (8% relatively). The impedance bandwidth of the measured array is 58.9–64 GHz (8.5% relatively).

Figure 12 compares the simulated and measured frequency responses of AR for the array with larger vias. Obviously, a better match with the simulations was obtained in this case as well. Nevertheless, the match stays poor. Considering the maximum manufacturing accuracy achievable in the university lab, a satisfactory agreement is impossible to be reached.

Since the dimensions of the coupling slots between the substrate layers do not influence the AR characteristics [14], the observed differences cannot be accounted for in those manufacturing imperfections.

The sensitivity analysis showed that the dimensions of the patches are the only parameters effectively influencing the AR characteristics [14]. Hence, a fit was searched by varying these dimensions. The closest fit was obtained for the situation where the width of the patch was increased by 0.1 mm while the length of the patch remained the same. The resulting response still showed only a limited match with the measured data.

The rest of the differences were expected to be accounted for by imperfect vias manufacturing, where not all the vias were correctly created, despite the enlarged diameter. Unfortunately, these effects were not feasible to be simulated due to the number of combinations of potential vias manufacturing errors.

The influence of manufacturing imperfections was investigated experimentally. In total, three samples of the 60 GHz antenna array with smaller vias and three samples with larger ones were manufactured with the same technology and the same accuracy. The variance of the parameters was significantly lower for the array with larger vias.

The AR bandwidth of the simulated array was from 56.5 GHz to 61.5 GHz (8.3% relatively). The AR bandwidth of the measured array was from 57.6 to 59.3 GHz (2.8% relatively).

Figure 13 shows a comparison of the radiation patterns for the simulated and manufactured arrays, with the larger vias diameter in the *xy* (up) and *yz* (down) planes. A good match with the simulated and measured characteristics was obtained for both planes. The beam width of the simulated array was 22° in the *xy* plane vs. 20° in case of the measured array. Beam width of the simulated array is 23° in the *yz* plane vs. 18° in the case of the measured array. Minor deviations of the main lobe direction were observed in both the *xy* and *yz* plane characteristics for the measured array.

The final dimensions of the array with larger vias are summarized in Table 1. Figure 14 shows how each parameter corresponds to the design of the array. Unless denoted otherwise in the figures, all dimensions related to the SIWs are referenced to the centers of the vias.

### 3.3. Additional Patches

Similar to the original array for the center frequency of 17 GHz, the converted array was also subsequently extended with parasitic patches to investigate their influence on the AR bandwidth of this structure [16,17]. As depicted in Figure 15, a matrix of parasitic patches of the same shapes, sizes, and orientation as the radiating patches was placed above the radiating patches at a defined distance. ARLON FoamClad with a thickness *t* = 1.88 mm and dielectric constant *ε_r_* = 1.25 was chosen as the carrier substrate. The distance between the parasitic and radiating patches was reduced to 0.8 mm. Additional fixing elements on the sides of the array were custom-designed and 3D-printed using a PETG filament.

In Figure 16, the simulated frequency response of AR for the array with parasitic patches stays on the level of 3 dB from 56 GHz to 68 GHz. When comparing these simulations with measurements of the arrays with parasitic patches and without patches, a surprising result is obtained: the parasitic structure does not make any visible effect on the characteristics. This effect is caused by the imperfect AR properties of the original array described in the previous section (added to the figure for convenience), which was used for the experiment and extended with the parasitic structure. Its inherent that sub-optimal AR properties are superimposed to the result. Hence, the use of parasitic patches makes sense only if a sufficient manufacturing accuracy can be ensured.

## 4. Summary

In the antenna lab, we were asked to develop a compact planar array for the operation in the 17 GHz unlicensed band. When developing the array, the following procedure was followed:A 2 × 2 patch array was selected as an initial structure to reach an initial gain and radiation properties. In order to allow an improvement of the gain and directive properties in the future, a modular concept was designed so that the array can be easily extended to 4 × 4, 8 × 8, etc. elements.Since the parasitic radiation of feeders becomes significant at 17 GHz, a substrate-integrated feeding network was designed. In the initial simulations, continuous side walls were assumed, and losses in the metallic components were neglected. The dimensions of the feeders, slots, and radiators (all related to the wavelength) were optimized.Continuous side walls were replaced by vias, and the numerical model was conceived to be as realistic as possible.

As a result, a good match between simulations and measurements was achieved, as presented in [4].

Since the design of the 17 GHz array was successful, the lab was asked to upscale the antenna for use in the 60 GHz 5G frequency band. When upscaling, the following procedure was followed:Since all dimensions of the radiators and feeders are related to the wavelength, a rough redesign considered wavelength shortening when thinner substrates at 60 GHz were used. For the initial optimization, a simplified numerical model with continuous side walls was used.Since the optimization of the axial ratio was not successful, a layer with parasitic patches was placed above the array. Thanks to the higher number of degrees of freedom, the subsequent optimization was successful.When the realistic model was developed and the first antenna prototype was produced, problems with manufacturing tolerances were revealed. Therefore, deep sensitivity and tolerance analyses were performed. Those analyses showed that the increased radius of vias in the side walls of SIW feeders might solve the problem. A new value of radii of vias was defined by the workshop, and the design was optimized afterwards accordingly.

As shown in the paper, conversion of the array to the higher center frequency was challenging from a manufacturing viewpoint. When converting, the following conclusions were formulated:The attainable manufacturing precision needs to be thoroughly considered and assessed. The sensitivity analysis indicated that a manufacturing precision with less than ±0.1 mm variance of the dimensions around the center (ideal) value is needed.The manufacturing of vias has to be considered as becoming generally more difficult in the case of the microwave substrates like ARLON CuClad 217 LX. Drilling such substrates, a residual material in drilled holes is left that complicates the metallization processes. This issue grows with the decreasing diameter of the vias. Hence, the diameter of the vias needs to be carefully chosen so that the acceptable manufacturing quality of the vias can be guaranteed.Since the width of the coupling slots between the substrates is reduced to 0.2 mm, precise alignment of the substrates is critical. It is desirable to include additional alignment points to the design in order to increase the precision.

In Table 2, we compare the differences in the dimensions and performances of the manufactured arrays operating at the center frequencies of 17 and 60 GHz. Obviously, the dimensions of the array operating at 60 GHz are much smaller due to the reduced wavelength that determines the sizes of most of the elements. The performance data of the array operating at 60 GHz show less balanced results, which is mainly the result of a lower match with the simulations caused by the manufacturing imperfections.

The measured results of the array extended with the parasitic structure differ from the simulations due to the inherited imperfections of the antenna array used for the experiment. In order to achieve a good performance of the parasitic structure, the optimal properties of the base antenna array are critical. The requirements for the manufacturing precision are therefore increased.

The modular properties of the design, inherent scalability, and potential for integration of advanced features such as beamforming make it a suitable solution for the cheap realization of fully customized and adaptive point-to-point links (indoors and outdoors) within the 5G networks in the 60 GHz band that can be integrated into the devices directly.

## Figures and Tables

**Figure 1 sensors-22-02945-f001:**
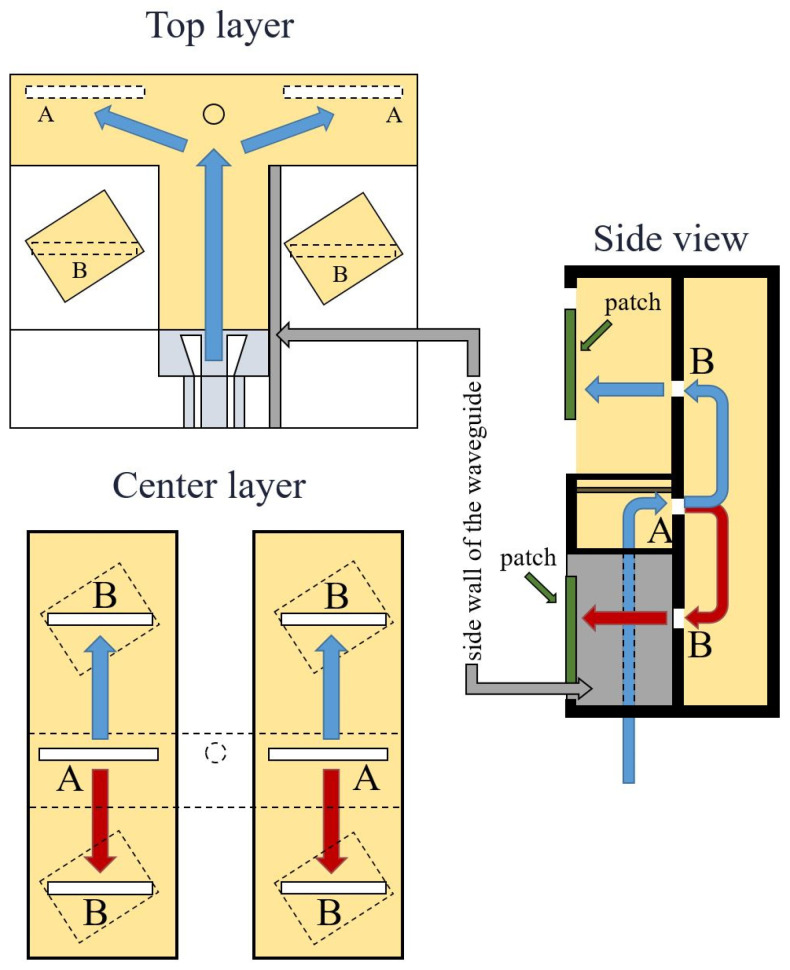
Concept of a two-layer 2 × 2 antenna array fed by substrate-integrated waveguides. Input power from grounded coplanar waveguide enters the top substrate via slots A and, through the bottom substrate, excites patches via apertures B.

**Figure 2 sensors-22-02945-f002:**
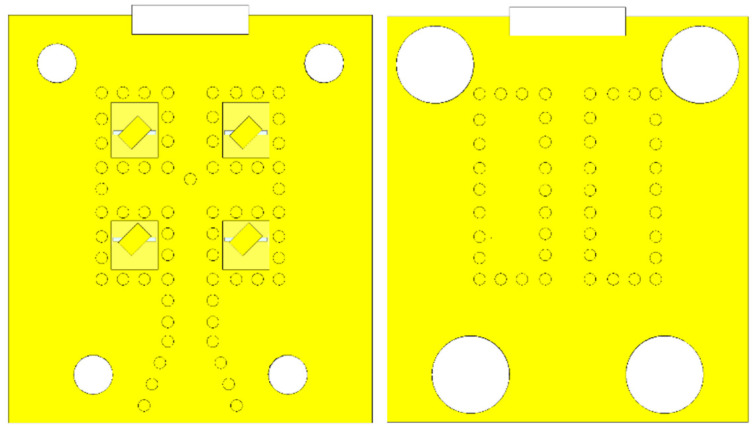
Positions of the SIW vias in the top layer (**left**) and the bottom layer (**right**).

**Figure 3 sensors-22-02945-f003:**
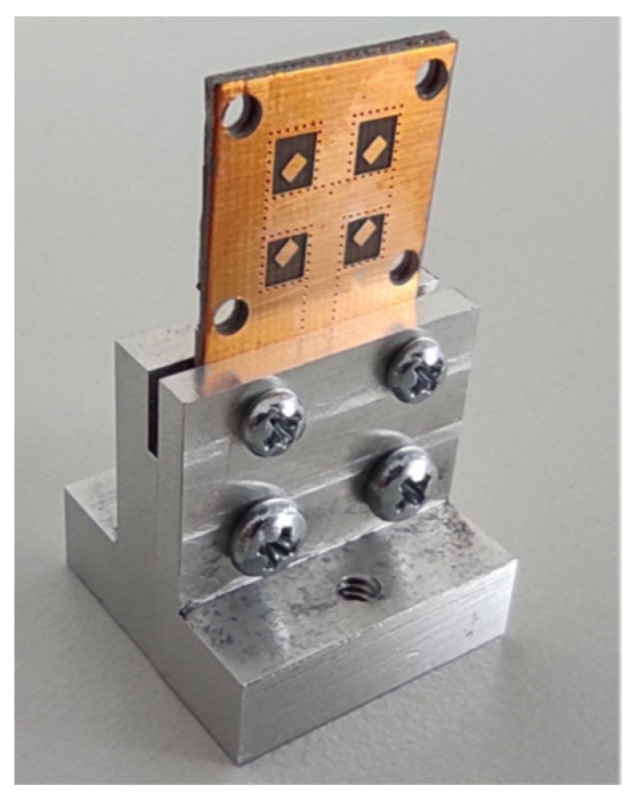
Manufactured array with a smaller vias.

**Figure 4 sensors-22-02945-f004:**
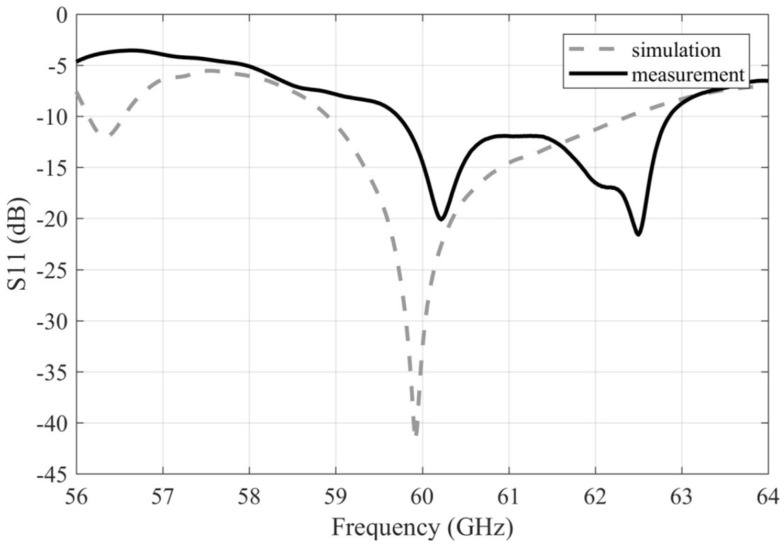
Frequency responses of the magnitude of the reflection coefficient for the simulated and measured antenna arrays with smaller vias.

**Figure 5 sensors-22-02945-f005:**
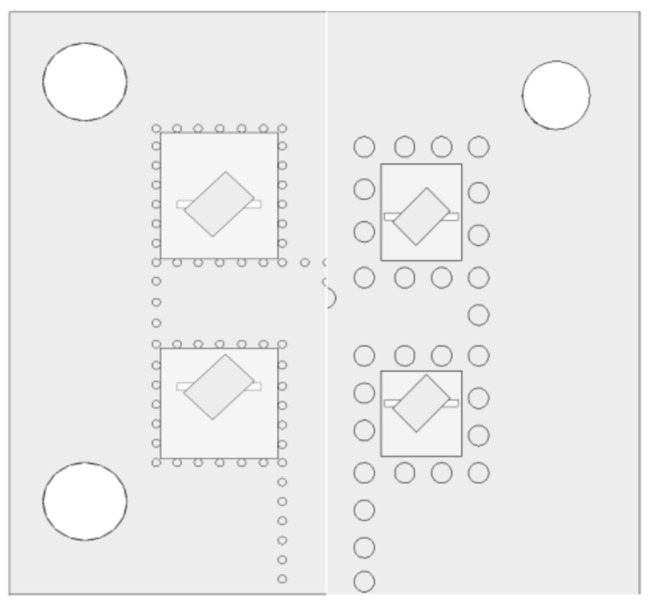
Comparison of arrays with smaller vias (**left**) and larger vias (**right**). Only half of the structures are shown due to symmetricity.

**Figure 6 sensors-22-02945-f006:**
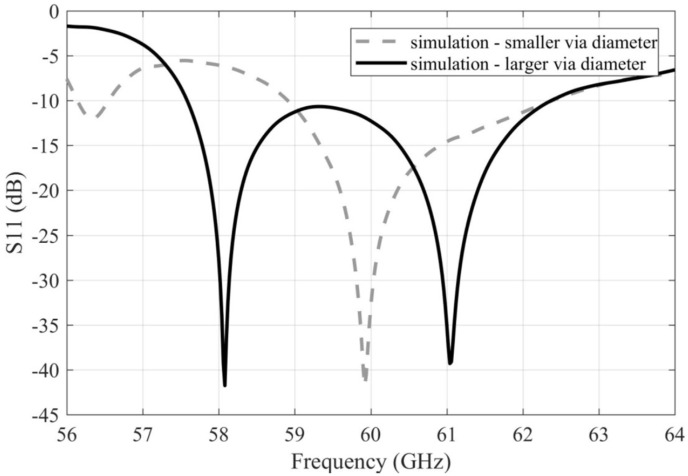
Frequency responses of the magnitude of the reflection coefficient for the simulated arrays with smaller and larger vias.

**Figure 7 sensors-22-02945-f007:**
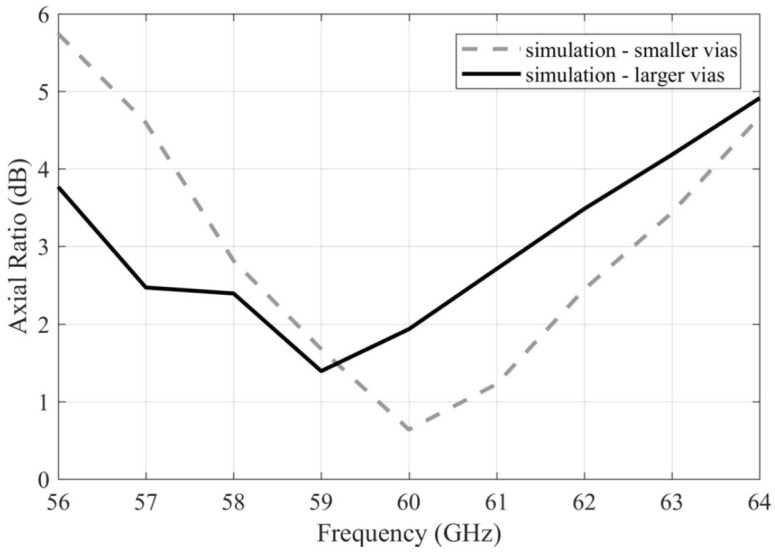
Frequency responses of the AR magnitude for the simulated arrays with smaller and larger vias.

**Figure 8 sensors-22-02945-f008:**
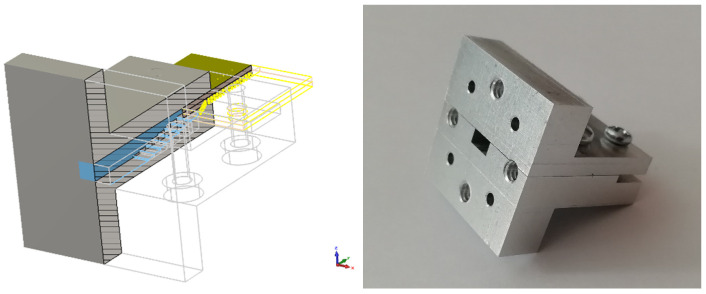
WR15 to SIW transition.

**Figure 9 sensors-22-02945-f009:**
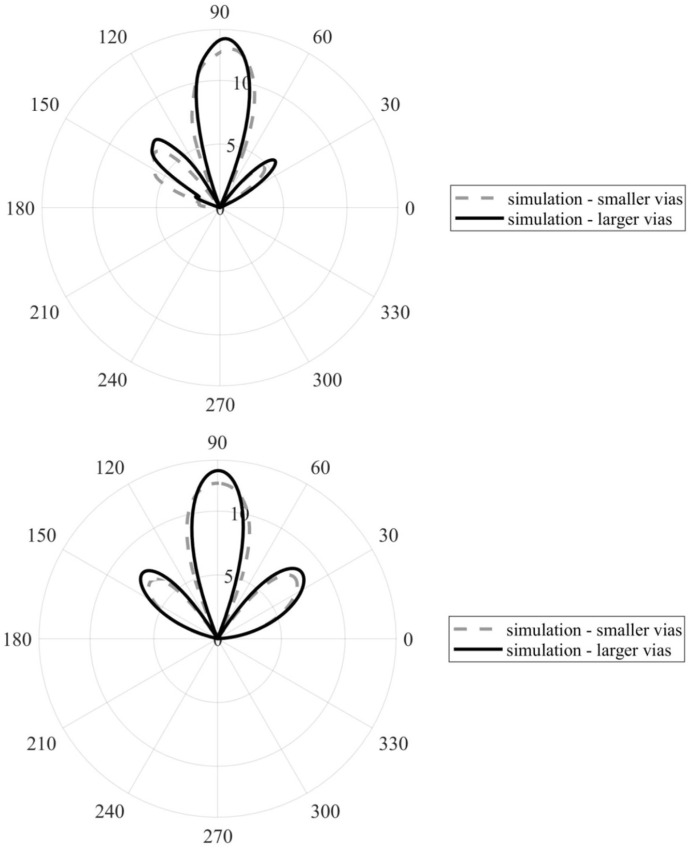
Comparison of the radiation patterns of the arrays with smaller and larger vias in the *xy* plane (**up**) and *yz* one (**bottom**). Shown in realized gain (dB).

**Figure 10 sensors-22-02945-f010:**
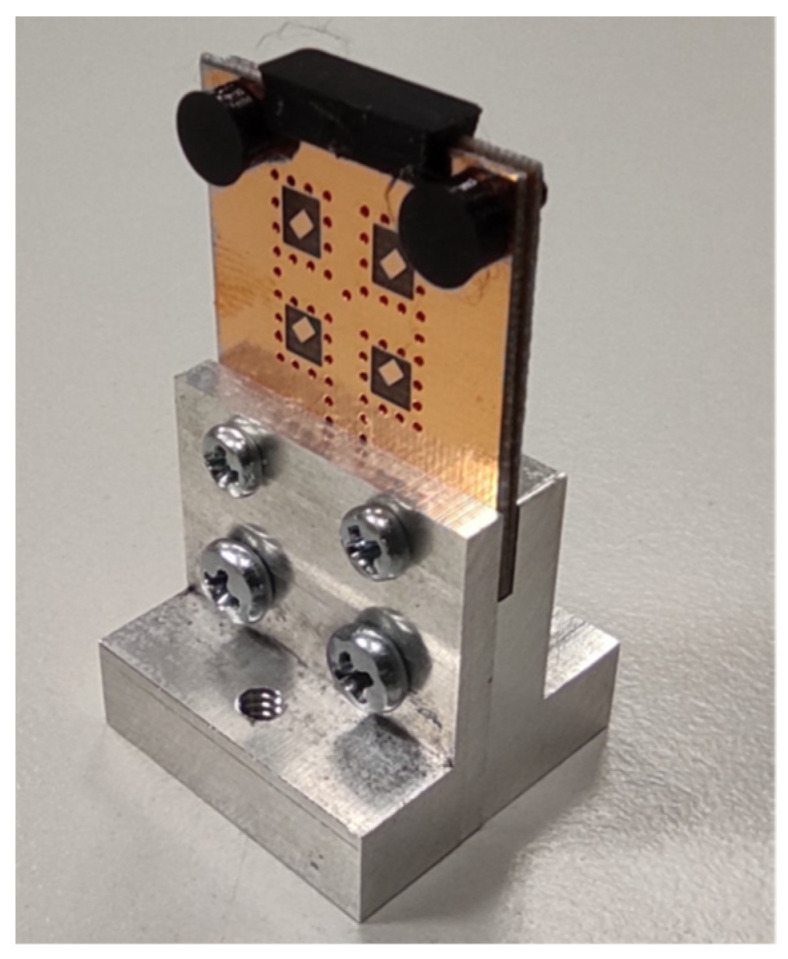
Manufactured array with the larger vias.

**Figure 11 sensors-22-02945-f011:**
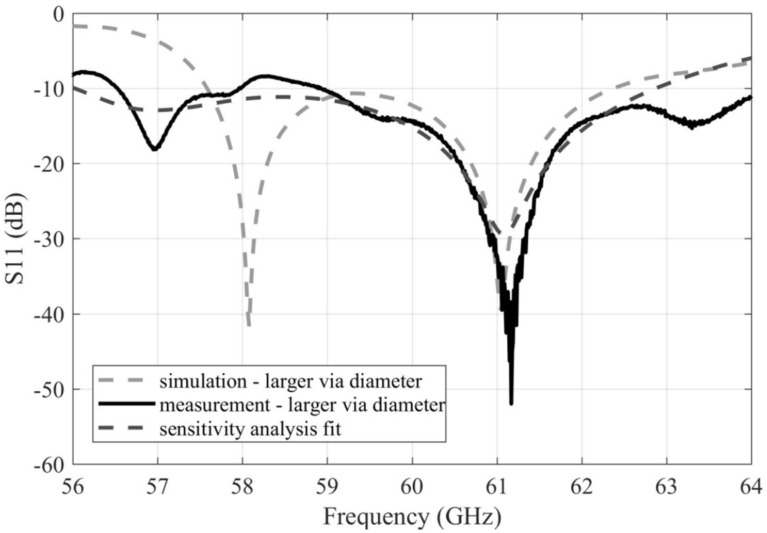
Frequency responses of the magnitude of the reflection coefficient for the simulated and measured arrays with the larger vias.

**Figure 12 sensors-22-02945-f012:**
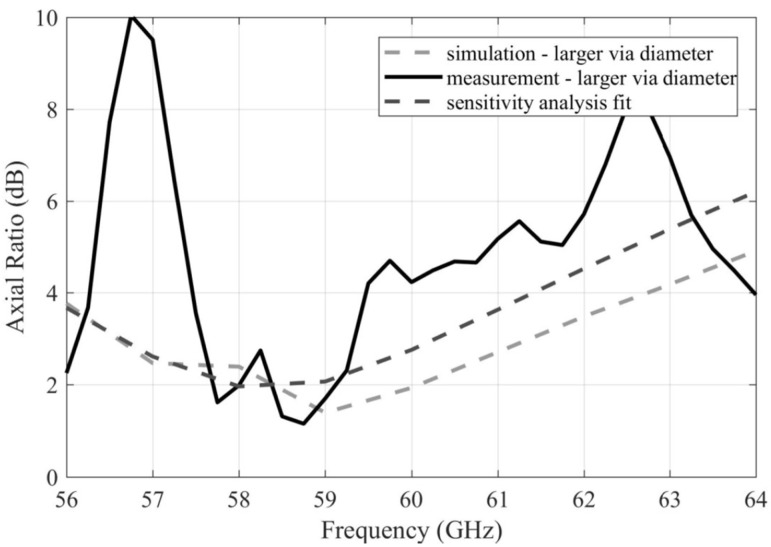
Frequency responses of the AR magnitude for the simulated and measured arrays with larger vias.

**Figure 13 sensors-22-02945-f013:**
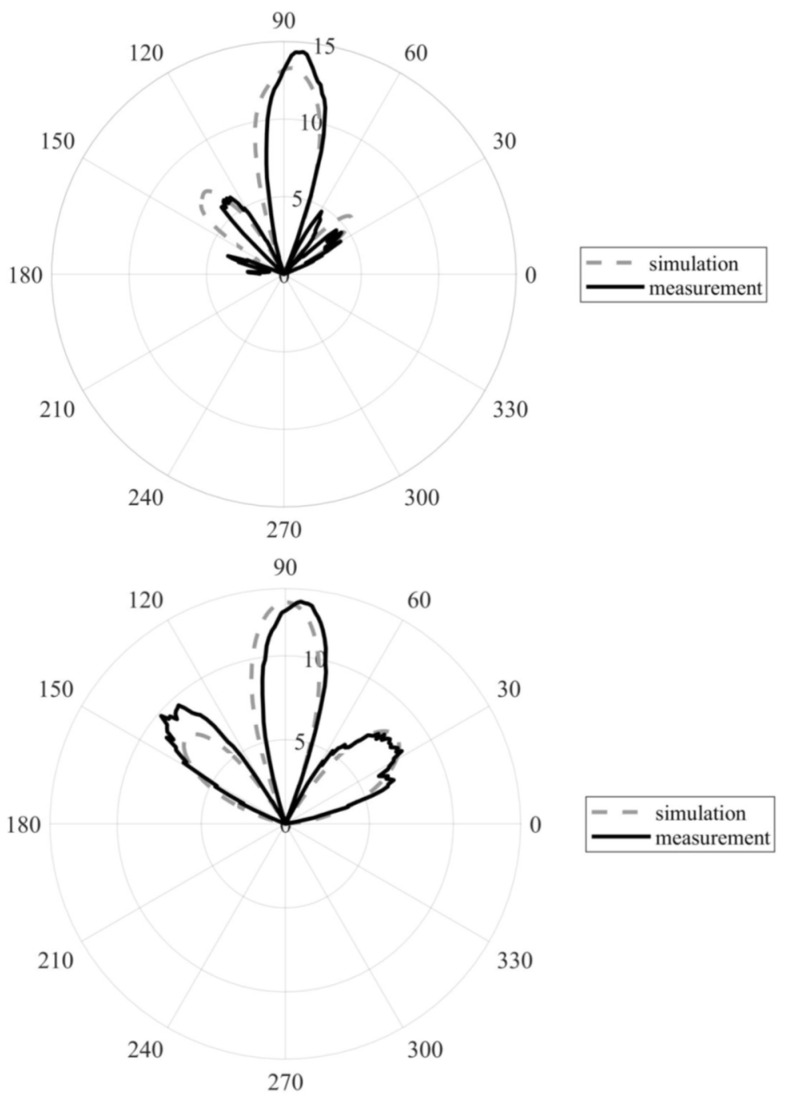
Comparison of the radiation patterns of the simulated and measured arrays with the larger vias in the *xy* plane (**top**) and *yz* plane (**bottom**). Shown in the realized gain (dB).

**Figure 14 sensors-22-02945-f014:**
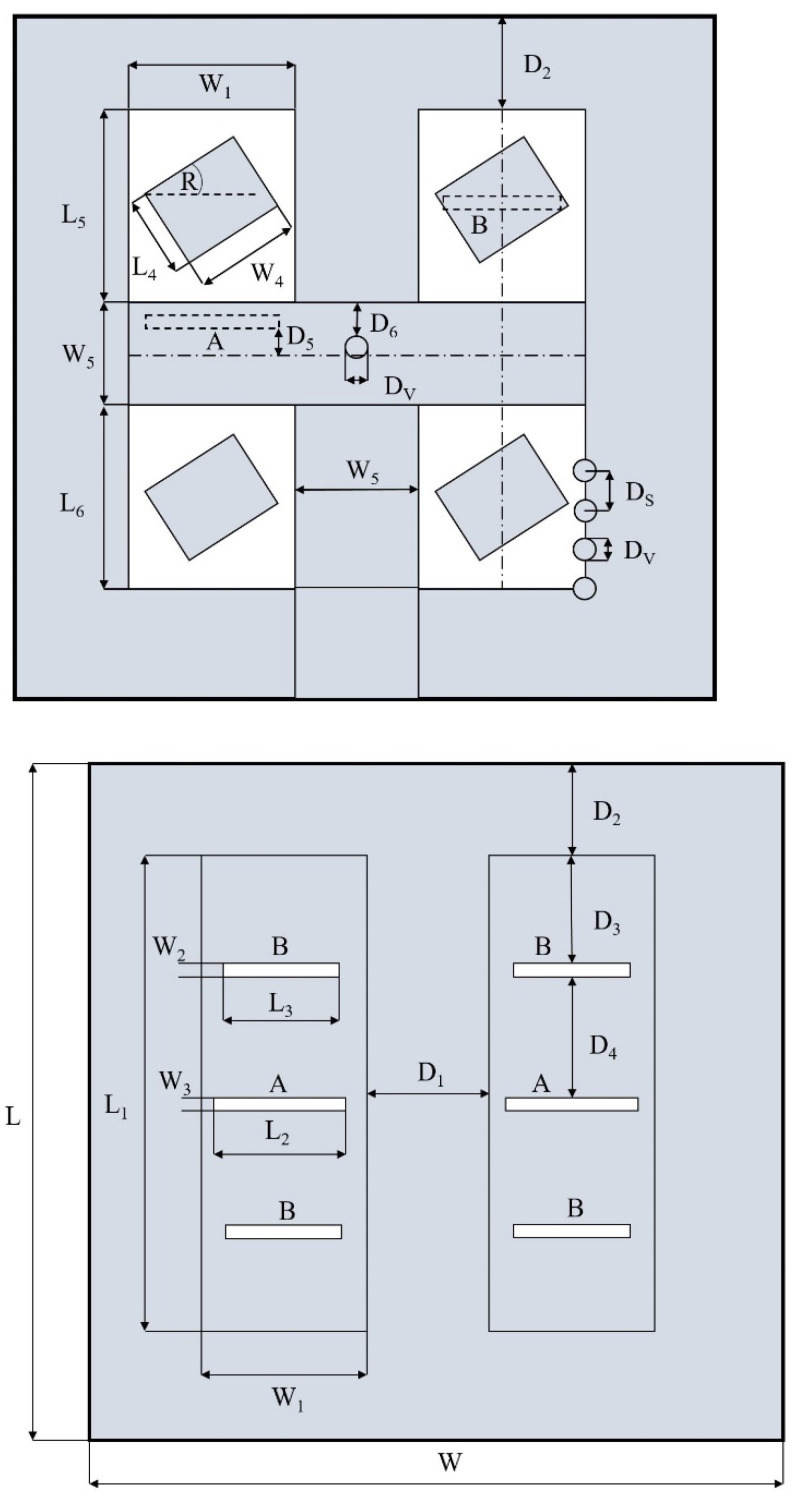
Dimensions of the array for the center frequency of 60 GHz for the top layer (**top**) and the bottom layer (**bottom**).

**Figure 15 sensors-22-02945-f015:**
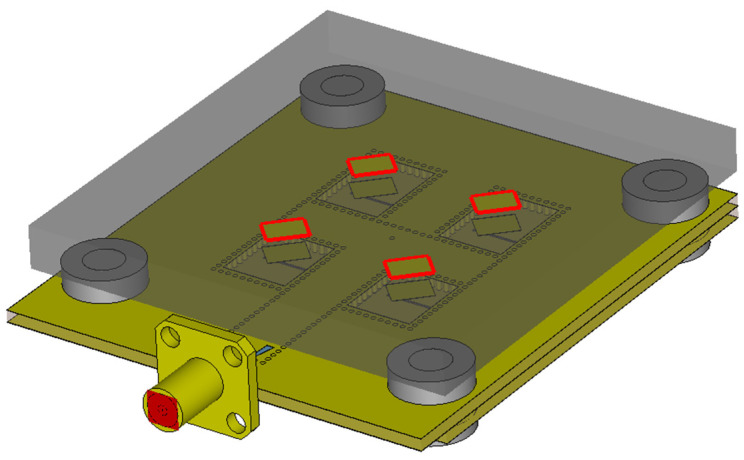
Antenna array with parasitic patches (bound in red).

**Figure 16 sensors-22-02945-f016:**
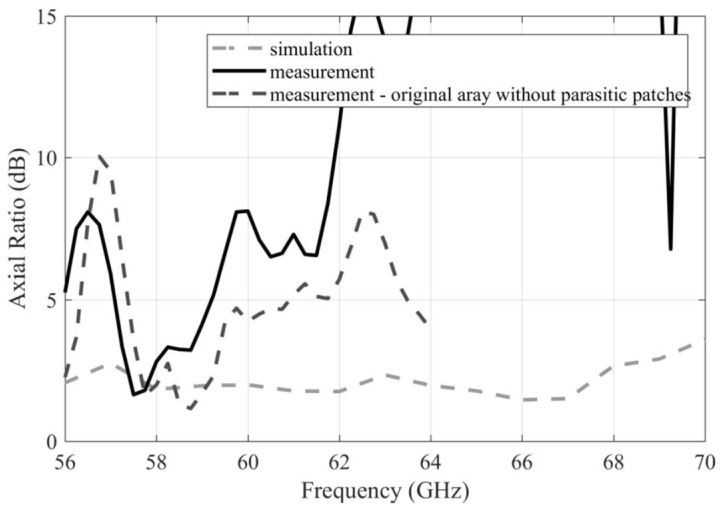
Frequency responses of the AR magnitude for the simulated and measured arrays extended with the parasitic patches.

**Table 1 sensors-22-02945-t001:** Dimensions of the array.

Symbols	Description	Value
L	length of the array	21.000 mm
L_1_	length of the SIWs in the bottom layer	9.600 mm
L_2_	length of the interlayer coupling slots	2.175 mm
L_3_	length of the patch coupling slots	2.200 mm
L_4_	length of the patches	0.984 mm
L_5_	length of the upper patch windows	3.850 mm
L_6_	length of the lower patch windows	3.450 mm
W	width of the array	18.732 mm
W_1_	width of the SIWs in the bottom layer	3.408 mm
W_2_	width of the patch coupling slots	0.200 mm
W_3_	width of the interlayer coupling slots	0.200 mm
W_4_	width of the patches	1.428 mm
W_5_	width of the SIWs in the top layer	2.316 mm
D_1_	spacing between the SIWs in the bottom layer	2.316 mm
D_2_	upper distance of the SIWs in the bottom layer from the edge of the array	4.000 mm
D_3_	distance of the patch coupling slots from the ends of the SIWs in the bottom layer	1.950 mm
D_4_	spacing between the patch and interlayer coupling slots	2.550 mm
D_5_	offset of the interlayer coupling slots from the center of the horizontal SIW in the top layer	0.016 mm
D_6_	distance of the divider via from the edge of the horizontal SIW in the top layer	0.300 mm
D_v_	diameter of the vias	0.600 mm
D_s_	spacing between the vias	1.150 mm
R	rotation of the patches	45°

**Table 2 sensors-22-02945-t002:** Comparison of dimensions and parameters of the arrays for the center frequencies of 17 and 60 GHz.

Symbols	Value (17 GHz)	Value (60 GHz)
L	65.000 mm	21.000 mm
L_1_	32.400 mm	9.600 mm
L_2_	7.500 mm	2.175 mm
L_3_	6.500 mm	2.200 mm
L_4_	3.920 mm	0.984 mm
L_5_	13.600 mm	3.850 mm
L_6_	11.600 mm	3.450 mm
W	57.270 mm	18.732 mm
W_1_	10.000 mm	3.408 mm
W_2_	0.300 mm	0.200 mm
W_3_	0.600 mm	0.200 mm
W_4_	5.080 mm	1.428 mm
W_5_	7.200 mm	2.316 mm
D_1_	7.200 mm	2.316 mm
D_2_	12.800 mm	4.000 mm
D_3_	7.250 mm	1.950 mm
D_4_	8.350 mm	2.550 mm
D_5_	0.700 mm	0.016 mm
D_6_	2.500 mm	0.300 mm
D_v_	0.600 mm	0.600 mm
D_s_	1.010 mm	1.150 mm
R	45°	45°
S11 bandwidth	4.8%	8.5%
AR bandwidth	7.1%	2.3%
Beamwidth (XY)	27.1°	20.0°
Beamwidth (YZ)	30.1°	18.0°
Realized gain (peak)	12.6 dB	14.3 dB

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
