# Peer review of "SIW-Based Circularly Polarized Antenna Array for 60 GHz 5G Band: Feasibility Study"

_sensors, 2022, doi:10.3390/s22082945_

Round 1
Reviewer 1 Report
The manuscript presents a circularly polarized antenna working at 60GHz. The results are nice but, however, I miss a much further explanation on the antenna itself since it is only provided an antenna working at 17 GHz and the extension to 60 GHz by optimizing a SIW structure. It seems that there are a lot of missed items to properly understand the methodology proposed by the authors.
The goal of the manuscript is not either clear since it provides different realizations and studies of the array (SIW with larger and smaller vias, parasitic patches) without leaving, from my point of view, with a clear design strategy.
Nonetheless I want to highlight the experimental work.
Finally, the English should be carefully reviewed since there are different typos that have to be corrected.
Author Response
See the enclosed document, please.

Reviewer 2 Report
-The paper presents a quite down-to-earth engineering work.
-Details of the CAD package (MoM, finite elements, 2D or 3D...) and how the structure was introduced in CAD tools would be quite interesting for readers (for example, is the metal base of the array introduced in the simulations? is the structure introduced completely or using symmetries? How unwanted substrate modes are prevented?) Maybe details like these could help in matching measurements and simulations)). Also some CAD tools have built in procedures for statistical analysis (Monte Carlo).
-Photos of the facilities used to obtain radiation patterns (anechoic chambers) would be quite illustrative.
-The las section 3.3 "additional patches" should be better illustrated.
Author Response
See the enclosed document, please.

Reviewer 3 Report
Based on previous papers bythe same authors (in particular [7]), the manuscript contains a description of a conversion (from 17 GHz to 60 GHz) of a small CP array. Since the authors have also realized and tested the antenna, the content is of interest.
However, the results are quite poor in general and, since nothing is said on the antenna requirements (even the required bandwidth for 5G at 60 GHz is not included nor thet antenna bandwidth is compared to it), it is impossible to evaluate whether the authors have reached their goal.
Moreover, the authors uses, for the 60 GHz antenna, the same laminate used at 17 GHz, a laminate which is an X-band one. I suspect that this material has high losses at 60 GHz. However, nothing is said about losses in the manuscript and, probably, many discrepancies between simulation and realization are due to the losses.
Moreover, this reviewer do not understand why authors have presented the straucture vith parasitic patches, since the authors themselves state (ro. 200-201) that these patches do not make any effect.
Further comments follow.
The comparisons (e.g., about Fig. 5 and 6, but there are others comparisons) are described but it would be better list them in tables, so that the reader can easily evaluate them. Then some words about the main point which can be derived from the tables can be commented.
The reference list contains an high percentage of papers from the same authors. Actually, all the cited papers are importnat, but a better analysis of the State-of-the-Art is required, which should lead to add several other references by other research group.
Fig. 1: Please, add the positons of the SIW vias (probably, a further subfigure is needed for this).
row 136: from which data you derive that statement?
row 147: "a better match..." is clearly better but still very poor.
Describe the WR15 to SIW transition.
row 158: a statistical analysis can be done.
Sect. 3.3: A better description of the added patches is needed. This reviewer has understood the structure only after looking at Fig. 6 of [8]
Author Response
See the enclosed document, please.

Round 2
Reviewer 1 Report
The paper has been technically improved and deserves to be published.
Nonetheless, the quality of English of the new writing is very poor and needs to be fully rewritten. It more seems an internal report than a potential journal paper.
Reviewer 3 Report
The revised version of the manuscript answers in a satisfactorily way many of the points raised by the reviewers.
However, a couple of points should be further improved. They follow.
1) Ref. [11] shows that the used laminate has a very good stability up to the Ku band, and probably the stability is rather good also at 60 GHz. However, a simple linear extrapolation of the data about ARLON CuClad 217LX shows that, at 60 GHz, thedielectric constant is probbaly 2.18 and, above all, the dissipation factor is at least around 0.002, i.e., twice the value at 10 GHz. But is probably even larger. So, I sugest to redo the simulations using these values, to check whether a better agreement can be reached. This also because the signal path in the antenna is rather long and, therefore, the losses can play a role.
2) In my previous review I've already said that the several references by the authors themselves are fully justified. But it seems strange to me that, on the topic of SIW antennas, and in particuar on SIW antennas at 60 GHz, no papers by others do exist. A good state of the art, including papers by other authors, on the paper topic (which is not how to increase the frequency of an antenna but, rather, how to realize a 60 GHz antenna, starting with antenna concepts which has been proved useful at X-band) is always a fundamental part of a publication. So, I suggest again to include a good state of the art in Sect. 1, and enrich the reference list accordingly.
Author Response
See the attached file, please.
